# Different Extraction Procedures Revealed the Anti-Proliferation Activity from Vegetable Semi-Purified Sources on Breast Cancer Cell Lines

**DOI:** 10.3390/antiox12061242

**Published:** 2023-06-09

**Authors:** Luigi Mandrich, Simona Piccolella, Antonia Valeria Esposito, Silvio Costa, Vincenzo Mercadante, Severina Pacifico, Emilia Caputo

**Affiliations:** 1Research Institute on Terrestrial Ecosystems-IRET-CNR, Via Pietro Castellino 111, 80131 Naples, Italy; 2Department for Environmental Biological and Pharmaceutical Sciences and Technologies, University of Campania “Luigi Vanvitelli”, Via Vivaldi 43, 81100 Caserta, Italy; simona.piccolella@unicampania.it (S.P.); severina.pacifico@unicampania.it (S.P.); 3Institute of Genetics and Biophysics-IGB-CNR, “A. Buzzati-Traverso”, Via Pietro Castellino 111, 80131 Naples, Italy

**Keywords:** breast cancer, drugs, glycerophosphocholine (GPC) derivatives, monogalactosyl-monoacylglycerol (MGMG), digalactosyl-monoacylglycerol (DGMG)

## Abstract

Breast cancer (BC) remains the leading cause of mortality in women, despite significant advancements in diagnosis. Thus, the identification of new compounds for its treatment is critical. Phytochemicals are known to exhibit anti-cancer properties. Here, we investigated the anti-proliferation potential of extracts from carrot, *Calendula officinalis* flower, and *Aloe vera* on breast cancer vs. epithelial cell lines. Various extraction methods were used, and the proliferative effect of the resulting extracts was assessed by proliferation assay on breast cancer and epithelial cell lines. Carrot, *Aloe* leaf, and *Calendula* flower extracts were extracted by hexane and methanol methods, and their semi-purified extracts were able to specifically inhibit the proliferation of breast cancer cell lines. The extract composition was investigated by colorimetric assays, UHPLC-HRMS, and MS/MS analysis. All the extracts contained monogalactosyl-monoacylglycerol (MGMG), while digalactosyl-monoacylglycerol (DGMG) and aloe-emodin were found in *Aloe*, and glycerophosphocholine (GPC) derivatives were identified in *Calendula*, except for the isomer 2 detected in carrot, suggesting that their observed different anti-proliferative properties may be associated with the different lipid compounds. Interestingly, *Calendula* extract was able to strongly inhibit the triple negative breast cancer MDA-MB-231 cell line proliferation (about 20% cell survival), supporting MGMG and GPC derivatives as potential drugs for this BC subtype treatment.

## 1. Introduction

Breast cancer (BC) is still the second largest cause of cancer-related mortality among women worldwide (AIRTUM-2018 https://www.registri-tumori.it/cms/, accessed on 30 April 2023), although significant advancements in its classification, diagnosis, and therapy have been developed [1,2]. Therefore, more treatment options are needed.

It is well known that the main sources of drugs are from minerals, animals, and vegetables [3]. However, despite several plant species being phytochemically studied, only a small percentage of them have been investigated for their pharmacological potential [4]. Therefore, many plant-derived chemicals with some pharmaceutical potential still remain to be discovered [5].

Moreover, it has been reported that the daily consumption of at least five different types of fruits and vegetables reduces the risk of developing a cancer by about the half [6,7]. Evidence has shown that the phytochemicals, found in fruits and vegetables, are the key compounds responsible for the chemo-protective mechanism [7]. For example, *Cruciferous* and *Brassicaceae* species have shown chemo-protective or anti-tumoral activity [8,9].

In this study, we investigated the potential anti-tumoral effect of extracts from carrot (*Apiaceae* family), *Calendula officinalis* flower (*Asteraceae* family), and *Aloe vera* leaf (*Asphodelaceae* family) on breast cancer and epithelial cell lines.

It has been reported that the oil extract from umbels of *Daucus carota* has anti-inflammation and anti-tumoral activity against human colon (HT-29, Caco-2) and breast (MCF 7, MDA-MB-231) cancer cell lines. It has been demonstrated that the oil extracted from dried umbels by a 50:50 acetone–methanol mixture has cytotoxic properties at a concentration of about 50 µg/mL [10]. In addition, it has been observed that the 6-methoxymellein extracted from carrots by methanol and purified on silica gel, inhibited the proliferation of breast cancer stem cells [11].

Furthermore, the *Calendula officinalis* flowers or “Marigold”, commonly known for their anti-inflammatory [12,13], antiviral [14], and genotoxic properties [15], have been reported to have a cytotoxic effect on tumor cell lines in vitro. It has been observed that the water extract of the flowers containing polysaccharides, proteins, carotenoids, acids, flavonoids, terpenoids, and saponins exhibited cytotoxic activity in vitro against several tumoral cell lines after its activation by laser [16]. In another study, the extract of the *C. arvensis* flowers, obtained by ethyl acetate, showed cytotoxic activity against MCF 7 and MDA-MB-231 breast cancer cell lines, with a value of IC_50_, in both cases, for about 80 µg/mL. Moreover, the analysis, by microscopy, of cells treated with this extract, showed chromatin condensation, nuclear fragmentation, and cell detachment, which are typical effects of cell death [17].

Finally, *Aloe vera* is among the most used herbals for its therapeutic properties [18,19] and it is also used as aliment [20]. Some of its derived compounds, such as Aloe-emodin, aloin, barbaloin, and aloesin, have shown potential anti-cancer activity [21,22,23,24].

Here, we reported a new extraction and a semi-purification method of compounds from carrot, *Calendula officinalis* flower, and *Aloe* leaf, which have been tested in cell proliferation assays on breast cancer cell lines. Moreover, both the crude extracts and the semi-purified fractions were analysed for their composition by using colorimetric assays and UHPLC-HRMS and MS/MS analysis.

## 2. Materials and Methods

### 2.1. Reagents

All chemicals were reagent grade from Sigma-Aldrich (USA). Columns were from GE Healthcare (USA).

### 2.2. Sample Preparation and Storage

*Aloe vera* and *Calendula officinalis* were cultivated without the use of chemicals and pesticides in an agricultural zone of the city of Acerra (near Naples, Italy). *Aloe vera* leaves and *Calendula officinalis* flowers were collected during the spring season, while carrots came from organic crops (Acerra, Italy). All the collected vegetables were completely dried under a stream of air at 45 °C and stored at 4 °C until their use (Figure 1).

### 2.3. Extract Preparation

A total of 2 gr of dried *Aloe vera* or *Calendula officinalis* were incubated in 10 mL of methanol for 6 h under stirring at room temperature, whereas 2 gr of dried carrot were incubated in 10 mL of hexane (6 h, RT). After incubation, the supernatants were recovered by centrifugation (10 min at 12,000 rpm, 4 °C), divided into aliquots of 2 mL, dried in a SpeedVac apparatus (Thermo Fisher Scientific, Waltham, MA, USA), and stored at −20 °C, as illustrated in Figure 1. The amount of extract recovered was between 40 and 50 mg for each aliquot.

### 2.4. Extract Purification 

The purification steps were the same for all the extracts. All following procedures were carried out at room temperature; in particular, a single dried extract (40–50 mg) was solubilized in 2 mL of methanol and loaded onto a Source 5RPC ST 4.6/150 column (GE Healthcare, USA; volume of the column was 2.5 mL, flow rate was 0.5 mL/min). The column was equilibrated with acetonitrile 10% (buffer A), washed with 10 mL of buffer A, and eluted with 60 mL of buffer B (acetonitrile 100%) using a linear gradient 0–100% B; the sample was recovered in 2 mL fractions and then tested in cell proliferation assay. The fractions showing activity were pooled and dried in a SpeedVAc apparatus (Thermo Fisher Scientific, USA). After dissolving in 2 mL of methanol, the pools were loaded onto a µRPC C2/C18 ST 4.6/100 column (GE Healthcare, Chicago, IL, USA; volume of the column was 1.5 mL, flow rate was 0.4 mL/min), which was previously equilibrated with buffer A. After washing (10 mL buffer A), samples were eluted with 40 mL of buffer B and recovered in 2 mL fractions. Fractions showing the highest proliferation inhibition were pooled and dried in a SpeedVac apparatus (Thermo Fisher Scientific, USA). The obtained samples were weighed and stored at −20 °C until their use. In Appendix A the chromatograms related to the purification of *Aloe vera*, *Calendula officinalis,* and *Daucus carota*, respectively, are reported.

### 2.5. Chemical Screening for Bioactive Substances

These analyses were performed to have an indication about the presence or not of some compound’s classes in the extracts and in their purified fractions. In particular, samples were investigated for the presence of saponins, tannins, alkaloids, steroids, terpenoids, flavonoids, and phenols.

Saponins. The presence of these compounds was investigated by the froth test, in accordance with Sholz et al. [25]; the formation of stable babbles indicates the presence of saponins.

Tannins. Tannins were detected by the gelatine-salt block test. Briefly, an aqueous 10% NaCl solution was added to an aliquot of the crude extracts or purified compounds, and the formation of a precipitate was reported [25,26].

Alkaloids. Samples were subjected to the Mayer’s test; only the samples presenting precipitates after the test were considered positive for the presence of alkaloids [25].

Steroids and terpenoids. Both classes of compounds were checked by the Salkowski test, in accordance with the colorimetric method of Gordon and Weber [27]. In the presence of steroids and terpenoids the solution changes from yellow to a red-orange color.

Flavonoids. The presence of both flavonoids and phenols was verified by using the Folin–Ciocalteu reagent [28], whereas only flavonoids were identified by using the Shinoda test: A fragment of magnesium ribbon and concentrated hydrochloric acid were added to the extracts. The appearance of a red to pink color indicates the presence of flavonoids [28].

Phenols. The presence of phenol/flavonoids was tested by the Folin–Ciocalteu reagent, and the presence of only phenols was tested by the ferric chloride test. Briefly, drop by drop a neutral ferric chloride (FeCl_3_) 2% solution was added to the extracts, and the formation of a green color indicates the presence of phenols [29].

The antioxidant activity was also assessed by means of the ABTS assay (2,2′-azino-bis (3-ethylbenzothiazoline-6-sulfonic acid)), which is a rapid, sensitive, and reproducible procedure [25,30]. The activity is revealed by the formation of a blue-green solution color.

### 2.6. UHPLC-HRMS and MS/MS Analysis of Semi-Purified Fractions

The ultra-high-performance liquid chromatography (UHPLC) method was carried out on a NEXERA UHPLC system (Shimadzu, Tokyo, Japan), using a Luna^®^ Omega C18 column (50 × 2.1 mm i.d., 1.6 μm particle size; Phenomenex, Torrance, CA, USA). The mobile phase consisted of water (solvent A) and acetonitrile (solvent B), both acidified with 0.1% formic acid. A linear elution gradient was optimized, starting from 45% B, and held for 0.5 min; then, it was ramped to 95% B in 10 min and was kept at that level for 2 min before restoring the initial conditions. The flow rate was 0.5 mL/min, and the injection volume was 2 µL.

High-resolution mass spectrometry analyses and tandem mass experiments (HRMS and HR-MS/MS) were performed by using the AB SCIEX TripleTOF^®^ 4600 spectrometer (AB Sciex, Concord, ON, Canada) equipped with a DuoSpray™ ion source operating in negative electrospray (ESI) ion mode. The APCI probe was used for automated mass calibration in all scan functions using the Calibrant Delivery System (CDS). An untargeted approach was developed, consisting of a full scan TOF survey in the mass range of 100–1300 Da with an accumulation time of 250 ms, and eight Information Dependent Acquisition (IDA) MS/MS scans in the mass range of 80–1100 Da with an accumulation time of 100 ms. Applied source parameters were the following: curtain gas (CUR) 35 psi, nebulizer gas (GS 1) 60 psi, heated gas (GS 2) 60 psi, ion spray voltage (ISVF) 4.5 kV, interface heater temperature (TEM) 600 °C, declustering potential (DP) 80 V, collision energy (CE) 45 V with a CE spread of 20 V. The instrument was controlled by Analyst^®^ TF 1.7 software, while data processing was carried out using PeakView^®^ software version 2.2.

### 2.7. Cell culture, Staining, and Proliferation Assay

Total and semi-purified extracts have been tested on human breast cancer cell lines, including MCF7 (human breast adenocarcinoma cell line, expressing estrogen receptor), SK-BR-3 (human breast adenocarcinoma cell line, expressing Her-2), and MDA-MB-231 (an epithelial, human breast cancer cell line, triple negative for estrogen, progesterone receptor, and Her-2), and on MCF10A (an epithelial, human breast cell line). All the cell lines were purchased by ATTC. All breast cancer cell lines were cultured in DMEM supplemented with 10% heat-inactivated fetal bovine serum (Life Technologies, Milan Italy), antibiotics, and glutamine. MCF10A cell line was cultured in Advanced DMEM/F12 supplemented with 5% horse serum, hydrocortisone (0.5 μg/mL), cholera toxin (100 ng/mL), insulin (10 μg/mL), antibiotics, and glutamine. Cell lines were tested, during experiments, for mycoplasma by using a Mycoplasma PCR Detection Kit (ABM-G238, Applied Biological Materials, Richmond, BC, Canada). All the cell lines were grown in 96-well plates (3000 cells/well), and after 24 h of incubation, different amounts of total or semi-purified extracts dissolved in DMSO were added. A final concentration of 1.0% DMSO was used, as a control. After 24, 48, 72, and 96 h of treatment, the cell viability was measured by using a CyQUANT^®^ Cell Proliferation Assay Kit (Invitrogen, Waltham, MA, USA). The better time to detect the effect was 48 h and, therefore, we used it as the treatment time in this study. Untreated and treated cell lines at different times were stained by crystal violet (1 mg/mL; Sigma-Aldrich, Burlington, MA, USA) and observed using a Leica (Wetzlar, Germany) 6000DMI microscope.

## 3. Results

### 3.1. Extract Preparation and Analysis of Their Biological Activity

Different extraction protocols from vegetables have been reported, and the resulting extracts have been used in cytotoxic, cell proliferation, and anti-inflammatory assays [10,16,17].

In this study, extracts from carrots, *Calendula* flowers, and *Aloe* gel and leaves have been prepared. After plant collection, washing, and drying, as illustrated in Figure 1, we performed three types of extraction procedures by using solvents with different polarities: phosphate buffer 10 mM, pH 7.0; hexane; and methanol.

In order to find which extraction procedure revealed the higher anti-proliferation activity on cancer cells, all the obtained crude extracts were dried, weighted, dissolved in DMSO at a final concentration of 10 mg/mL, and tested by cell proliferation assays. As a cell system, we used the breast cancer cell line MDA-MB-231, and as control we used the MCF10A cell line. Previously, DMSO toxicity was tested at different concentrations (0.25–0.5–1.0–2.5–5.0–7.5% *v*/*v*) in our cell system. The 1.0% *v*/*v* DMSO final concentration seemed not to affect the number and the morphology of the examined cells (Figure 2), and thus, it has been used in all the experiments.

The extracts were used in proliferation assays at 4 different concentrations: 0.5–5.0–50–250 µg/mL. Cell survival was evaluated upon 48 h cell treatment, with the different extracts, by CyQUANT^®^ Cell Proliferation Assay (Figure 3).

For carrot extract, we observed that the best extraction method was by using hexane (Figure 3). In fact, the obtained extract was able, at a final concentration of 0.5 µg/mL, to reduce the MDA-MB-231 cell survival by about 50%, and this value decreased to 20% survival by increasing the extract concentration up to 250 µg/mL, whereas all extract concentrations, when tested on MCF10A breast epithelial cell line, did not affect their cell survival. This finding suggested that the effect was specific only to cancer cells. The carrot extract in phosphate buffer did not influence the survival of either cell line at all concentrations examined (Figure 3A), while the methanol extract showed a weak effect only at a concentration of 250 µg/mL on the breast cancer cell line (Figure 3C). Based on these data, carrot extract by hexane has been used in all the experiments reported in this study.

In the case of *Calendula officinalis* flowers, the best extraction method was in methanol (Figure 4); in fact, cancer cell survival was of about 40% when the cells were treated with 0.5 µg/mL of this extract, whereas breast normal cells showed a survival among 80–100% at all extract concentrations tested.

The *Calendula officinalis* flowers extract in phosphate buffer did not show an effect on cell survival at all concentrations tested (Figure 4A). Whereas hexane extract at 5.0, 50, and 250 µg/mL concentrations showed a slightly and similar effect on the cancer cell survival that was reduced to about 30% compared to the no-treated cancer cells (Figure 4B). Based on these results, the method of extraction by methanol has been used for *Calendula officinalis* flowers in all the experiments reported.

In the case of *Aloe vera,* proliferation assays have been performed using the gel and leaf separately, since it has been reported that only the gel exhibited cytotoxic activity [31]. The gel was dried, and its extract prepared by the three different extraction procedures, as described in the Materials and Methods. We found that gel extracts from all three extraction procedures showed a comparable cytotoxic activity on both, normal and cancer, cell lines (Figure 5).

Differently, *Aloe* leaf extracts from the methanol procedure decreased the cell survival of about 50% in cancer cells at 5.0 µg/mL, as shown in Figure 6, and a very weak effect, only at a 50–250 µg/mL concentration range on the breast epithelial cell line. This extraction procedure was then used for all subsequent experiments.

The MCF10A and MDA-MB-231 cell lines, non-treated and treated with the different extracts, were fixed and stained by crystal violet, as described in the Materials and Methods and visualized by microscope. As shown in Figure 7, the breast cancer cell line after treatment with the different extracts were reduced in number, while the MCF10A number was unchanged upon treatment, as previously observed.

### 3.2. Extract Semi-Purification, Colorimetric Assays, and Chemical Profiling by UHPLC-HRMS

With the aim of obtaining more homogeneous and clean samples, after the identification of the best extracting conditions for our three vegetables, the extracts were partially purified by carrying out two steps on hydrophobic chromatographic columns, having different capacity and hydrophobicity. All different extracts were loaded onto a Source 5RPC ST 4.6/150 column (see Materials), eluted in 2 mL fractions, and tested for their cytotoxic activity. We observed that the fractions containing biological active compounds were eluted from all three vegetables, between 70 and 100% acetonitrile (Appendix A). These fractions were then pooled, dried, and used in a second purification step by using a µRPC C2/C18 ST 4.6/100 column (see Materials). The resulting fractions were eluted in a 2 mL volume and tested for their cytotoxic activity. We found that the fractions, obtained from all three vegetables, containing biologically active compounds, were the ones eluted with around 90% acetonitrile (Appendix A). These fractions were pooled, dried, and indicated as semi-purified extracts.

To gather information about the compounds present in the extracts and in their final semi-purified fractions from carrots, *Calendula officinalis* flowers, and *Aloe vera* leaves, colorimetric assays were performed. In particular, we analysed the presence in these samples of the following eight possible compound classes: saponins, tannins, antioxidants compounds, alkaloids, steroids, terpenoids, flavonoids, and phenols. As shown in Table 1, we identified antioxidant compounds, alkaloids, and phenols, in the crude carrot extract, and only antioxidant compounds in its final semi-purified fractions. Instead, the crude extract from *Calendula officinalis* flowers contained alkaloids, steroids, terpenoids, and flavonoids, while in their final semi-purified fractions, only antioxidant compounds were identified. Finally, while *Aloe vera* leaf crude extracts showed the same classes of compounds of *Calendula* flowers, their final semi-purified fractions contained alkaloids and flavonoids.

Furthermore, to explore the chemical composition of the semi-purified extracts from the three plant species under study in greater depth, an untargeted approach was carried out by coupling UHPLC procedures with high-resolution tandem mass spectrometry, recorded in negative ion mode. TOF-MS and MS/MS data of compounds tentatively identified are summarized in Table 2, whereas a brief discussion of the evidence for compound putative identification is provided below.

Calendula sample was mainly composed of glycerophosphocholine (GPC) derivatives, differently substituted. Their molecular ions shared even *m*/*z* values in accordance with the presence of a nitrogen atom and appeared in the form of formate adducts ([M + FA]^−^). The first fragmentation, detected in ToF-MS/MS spectra, arose from the neutral loss of methyl formate (60 Da). This behavior has been described as pivotal for the identification of the GPC lipid class and seems to involve a gas-phase S_N_2-like reaction, in which the formate ion acts as the nucleophile and the methyl group of the quaternary ammonium polar head as the electrophile [32]. Therefore, a demethylated GPC deprotonated ion was generated, which, in turn, fragmented releasing its own fatty acid through the neutral loss of 225 Da and the corresponding product ion at *m*/*z* 224.069 (Appendix A). In particular, fragments at *m*/*z* 277 could be attributed to an 18:2 fatty acid (e.g., α-linolenic or calendic acid), while the one at *m*/*z* 295 could be ascribed to its hydroxy-derivative, already described in plants belonging to *Calendula* genus [33]. The latter was also found in free form in both calendula and carrot samples. The other polyunsaturated fatty acids (PUFAs) esterified to the glycerol skeleton were likely linoleic and/or linoelaidic acid (18:2; at *m*/*z* 279), oleic acid (18:1; at *m*/*z* 281), and saturated palmitic acid (16:0; at *m*/*z* 255).

Lipid Maps^®^ Structure Database (LMSD) [34] was consulted, when necessary, to check for chemical structures.

Apart from GPCs, one monogalactosyl-monoacylglycerol (MGMG) was tentatively identified in all the samples, whereas a DGMG (16:0; at *m*/*z* 699.3803) was found only in *Aloe* semi-purified extract. Additionally, these metabolites appeared as formate adducts, which in MS/MS experiments immediately gave rise to deprotonated structures. Then, ions at *m*/*z* 415.1470, assigned to the digalactosylglycerol moiety, and the corresponding dehydrated fragment at *m*/*z* 397.1363, were pivotal for their characterization.

Finally, in *Aloe* semi-purified extract, the metabolite with the molecular formula C_15_H_10_O_5_ (at *m*/*z* 269.0453) was identified as aloe-emodin according to the literature [35].

### 3.3. Semi-Purified Extract Biological Activity

Semi-purified extracts from *Calendula*, *Aloe*, and carrot were dried, weighted, solubilized in DMSO at a final concentration of 1 mg/mL, and tested at the final concentration of 0.5 µg/mL (assay concentration of DMSO 1.0% *v*/*v*). These semi-purified extracts were tested on four different cell lines: MCF10A as breast cell line of control and MCF7, SKBR3 and MDA-MB-231 breast cancer cell lines, representing oestrogen receptor positive, and Her-2 positive and triple negative breast cancer subtypes. All cell lines were treated upon 48 h with all the semi-purified extracts, and cell survival was evaluated by CYQUANT^®^ cell proliferation assay.

As shown in Figure 8, all the cell lines used showed a cell survival of about 100% when treated with 1.0% *v*/*v* of DMSO; the carrot semi-purified extract (the compounds present and their percentage composition are reported in Table 2) permitted 55, 40, and 50% cell survival, respectively, on MCF 7, SK-BR-3, and MDA-MB-231 cells; the *Calendula* semi-purified extract was very active (the compounds present and their percentage composition are reported in Table 2), showing a cell survival of about 50% in MCF 7 cells and of about 35 and 20% cell survival in SK-BR-3 and MDA-MB-231, respectively; *Aloe* leaves semi-purified extract resulted in a less efficient effect on MCF 7 cells showing about 80% survival (the compounds present and their percentage composition are reported in Table 2), whereas in SK-BR-3 and MDA-MB-231 cells survival was reduced to 40%.

## 4. Discussion

In this study, we investigated the ability of extracts from carrot, *Calendula officinalis* flower, and *Aloe vera* leaf to inhibit cell proliferation by using breast cancer as testing cell lines, being that this type of cancer is one of the most diffused and is characterized by a high level of mortality.

The first step was to set up an extraction procedure to isolate biologically active compounds from all of them. We found that the best extraction method able to reveal the presence of compounds with anti-proliferative properties was by hexane in the case of carrots, and by methanol for *Calendula* flowers. Our finding was different from the results reported about biologically active extracts from *Calendula* flowers obtained by water extraction [16] and from carrots by other organic solvents [10].

Moreover, *Aloe vera* was analysed separately, as gel and leaf, showing cytotoxic activity on both, normal and cancer cells as gel, and specific anti-proliferative activity as leaf extract. Despite *Aloe* gel being used in cosmetics and beverages, there is evidence that, at low concentrations of some compounds such as aloin, it shows anti-bacterial and antioxidant activities [36] as well as cancer cell apoptosis by modulating the mitochondrial metabolism [31]; while it has been reported to be toxic at high concentrations of components such as aloin [36,37]. These data are in agreement with the large toxicity of Aloe gel we observed on all the cell lines examined in this study (Figure 5).

Further, crude extracts and final semi-purified fractions were analysed to identify which compound classes were present; antioxidants, alkaloids, steroids, terpenoids, flavonoids, and phenols were found in crude extracts, while, in the final semi-purified samples, antioxidants were detected from carrot and *Calendula* flowers, and alkaloids and flavonoids from the *Aloe* leaf (Table 1).

UHPLC-HRMS and MS/MS analysis of semi-purified extracts from the vegetables under study, identified mainly glycerophosphocholine (GPC) derivatives, differently substituted in *Calendula* flowers except for GPC (16:0) isomer 2 as well as the hydroxy-octadecadienoic acid compound also detected in carrot. Instead, all the extracts contained one monogalactosyl-monoacylglycerol (MGMG, 18:3; at *m*/*z* 277.2174), while a digalactosyl-monoacylglycerol (DGMG; 16:0; at *m*/*z* 699.3803) and aloe-emodin were only identified in *Aloe* leaves semi-purified extract (in Table 2 the percentage composition of each sample are reported).

All together, these data suggest that the anti-proliferative effect may be associated with the presence of the glycoglycerolipid compound (monogalactosyl-monoacylglycerol, MGMG, 18:3; at *m*/*z* 277.2174) detected in the extracts from all vegetables. Previously, it has been reported that the monogalactosyldiacylglycerol (MGDG), another type of glycoglycerolipid found in the plant cell membrane, showed therapeutic anti-cancer activity [38]. The different levels of anti-proliferative activity observed in this study may be associated with the other compounds present in the different extracts. Interestingly, the anti-proliferative effect of the semi-purified extracts from *Calendula* flowers was higher, compared to the other extracts from carrot and *Aloe*, on the MDA-MB231 breast cancer cell line. This effect may be due to the presence of the different glycerophosphocholine (GPC) derivatives detected only in *Calendula* flowers, except for the GPC (16:0) isomer 2 which was also found in carrots.

## 5. Conclusions

Lipid compounds are getting attention for their anti-tumoral, anti-inflammatory, and anti-viral activities. It has been reported that they are able to induce greater apoptosis in various cancer cells (cervical, gastric, and prostate cancers) than in normal cells, both in vitro and in vivo, by generating reactive oxygen species, suppressing angiogenesis via the Akt cell signalling pathway, and inducing apoptosis [39,40,41,42]. We identified mainly lipid compounds in all three vegetable extracts used in this study. Interestingly, among the semi-purified extracts, the one derived from *Calendula* was able to strongly inhibit triple negative breast cancer MDA-MB-231 cell line proliferation among all the examined breast cancer cell lines (about 20% cell survival after 48 h treatment). BC includes several distinct subtypes based only on genetic, morphologic, and clinical features [43]. Among all of them, triple negative breast cancer (TNBC) is a long-lasting orphan disease, because of its poor prognosis and limited treatment options. Our finding suggests the potential use of this extract as drugs for the treatment of this BC subtype. However, further studies investigating the mechanisms of actions, intracellular uptake, as well as a screening involving additional cancer cell lines are needed to confirm the potential novel therapeutic efficacy of the identified lipid compounds for breast cancer treatment.

## Figures and Tables

**Figure 1 antioxidants-12-01242-f001:**
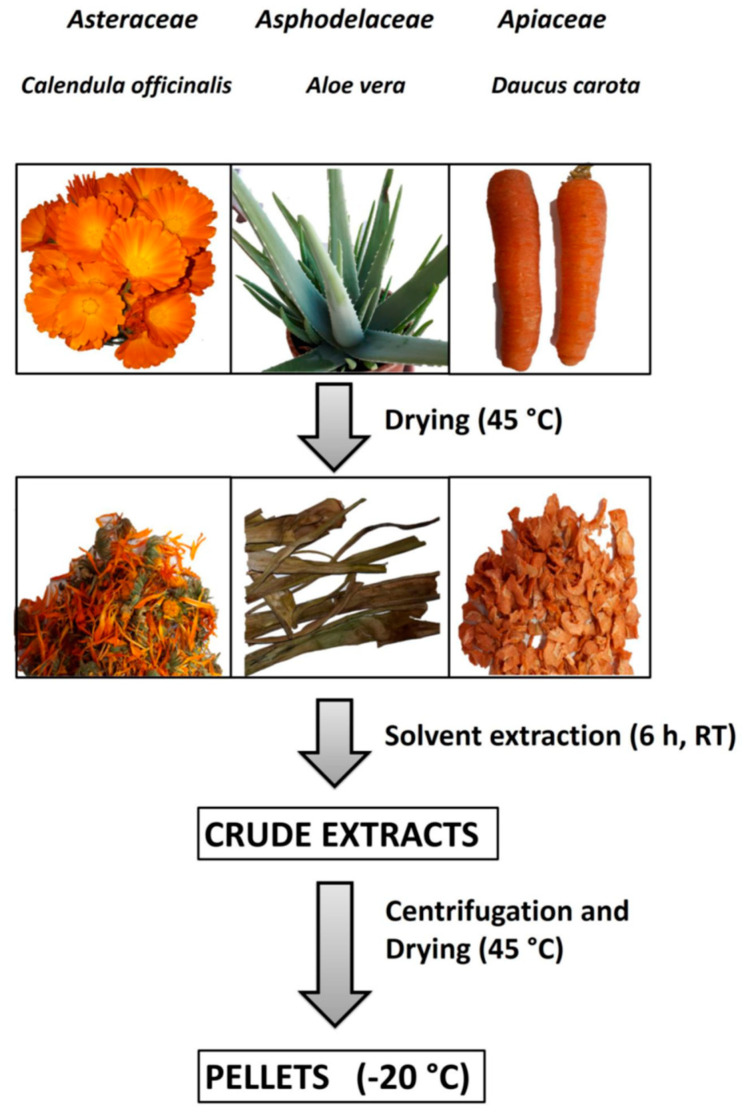
Vegetables preparation, extraction, and storage. The steps of sample preparation were plant collection, washing, and drying; solvent extraction; centrifugation and supernatant drying; and storage at −20 °C of dried samples.

**Figure 2 antioxidants-12-01242-f002:**
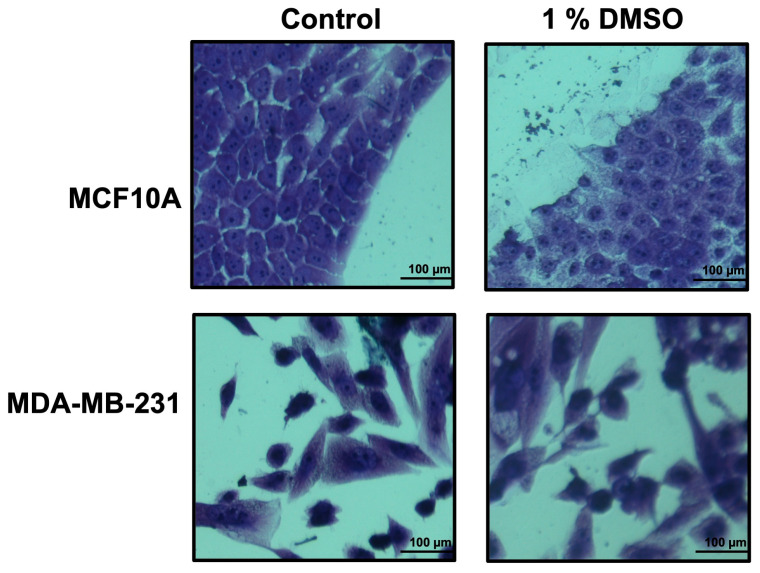
Growth of breast epithelial and cancer cell lines in medium containing DMSO 1%. Cell lines, during their growth in medium with/without DMSO 1%, were stained by using crystal violet and observed by microscope (magnification, ×20) after 48 h. The scale bar (100 µm) is shown.

**Figure 3 antioxidants-12-01242-f003:**
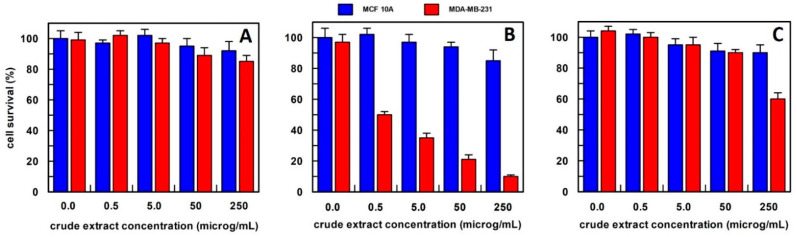
The anti-proliferative effect on breast epithelial (MCF 10A) and cancer cells (MDA-MB-231) of total carrot extract, obtained (**A**) in 10 mM buffer phosphate pH 7.0; (**B**) in hexane; and (**C**) in methanol. Results represent the means (percentage values) of 3 independent experiments performed in triplicate. The error bars show 95% confidence intervals.

**Figure 4 antioxidants-12-01242-f004:**
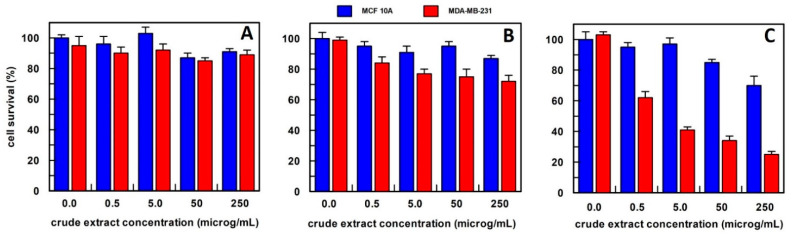
The anti-proliferative effect on breast epithelial (MCF 10A) and cancer cells (MDA-MB-231) of total *Calendula officinalis* flowers extract, obtained (**A**) in 10 mM buffer phosphate, pH 7.0; (**B**) in hexane; and (**C**) in methanol. Each point is the mean of the results (percentage values) of 3 independent experiments performed in triplicate. The error bars show 95% confidence intervals.

**Figure 5 antioxidants-12-01242-f005:**
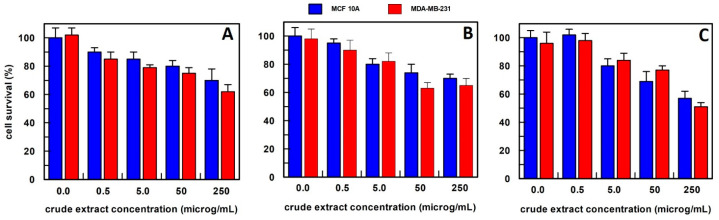
The anti-proliferative effect on breast epithelial (MCF 10A) and cancer cells (MDA-MB-231) of *Aloe vera* gel extract, obtained (**A**) in 10 mM buffer phosphate, pH 7.0; (**B**) in hexane; and (**C**) in methanol. Results represent the means (percentage values) of 3 independent experiments performed in triplicate. The error bars show 95% confidence intervals.

**Figure 6 antioxidants-12-01242-f006:**
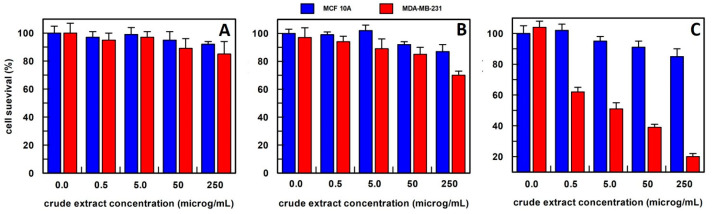
The anti-proliferative effect on breast epithelial (MCF 10A) and cancer cells (MDA-MB-231) of *Aloe vera* leaves extract, obtained (**A**) in 10 mM buffer phosphate, pH 7.0; (**B**) in hexane; and (**C**) in methanol. Results represent the means (percentage values) of 3 independent experiments performed in triplicate. The error bars show 95% confidence intervals.

**Figure 7 antioxidants-12-01242-f007:**
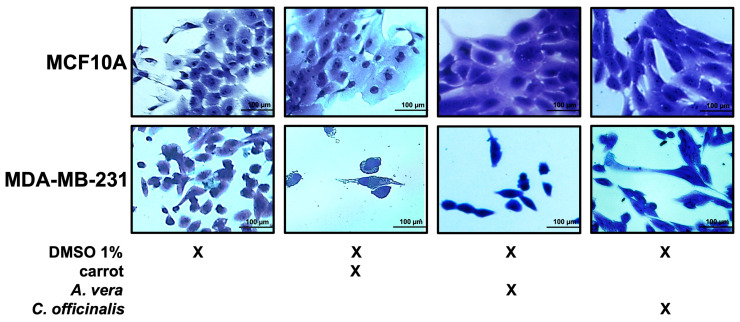
Growth of breast epithelial and cancer cell lines in the presence of the different extracts. Representative images of both cell lines during their growth in medium with DMSO 1%, or carrot, *A. vera*, or *C. officinalis* extract, each one used at a concentration of 0.5 μg/mL. Cells were stained with crystal violet and observed by microscope (magnification, ×20). The scale bar (100 µm) is shown.

**Figure 8 antioxidants-12-01242-f008:**
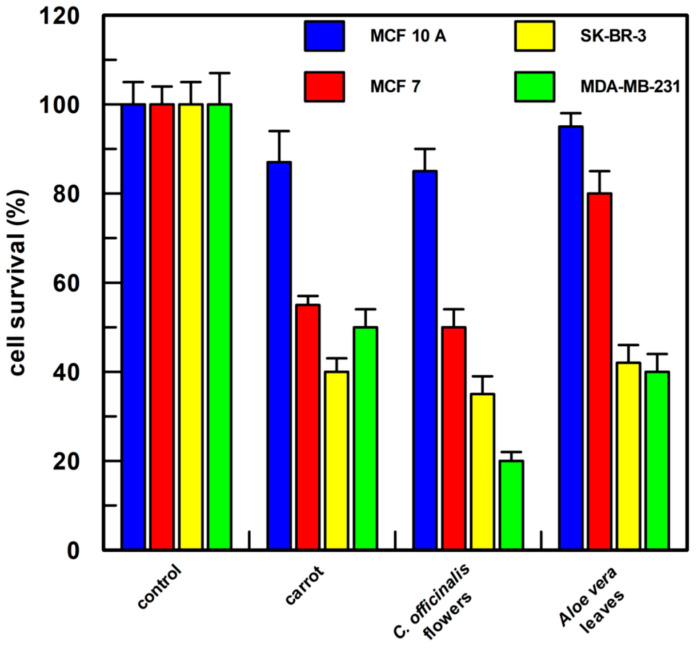
Cell proliferation assay. Each semi-purified extract was tested at a final concentration of 0.5 µg/mL in 1.0 % *v*/*v* of DMSO. After 48 h of treatment, cell proliferation was measured by CyQUANT^®^ Cell Proliferation Assay Kit (the compounds present in each sample and their percentage composition are reported in Table 2). MCF 10A, epithelial breast cell line, MCF 7, SK-BR-3, and MDA-MB-231 breast cancer cell lines were examined. Each point was measured in triplicate, and the result is the mean (percentage values) of the three measures; the error bars show 95% confidence intervals. The “control” was cells incubated only with 1.0% DMSO *v*/*v*.

**Table 1 antioxidants-12-01242-t001:** Colorimetric assays. Crude extracts and final purified active fractions from carrots, *Calendula officinalis* flowers, and *Aloe vera* leaves were analysed for the presence of eight classes of compounds: saponins, tannins, antioxidant compounds, alkaloids, steroids, terpenoids, phenols, and flavonoids (for assays details see Materials and Methods).

Vegetables	Purification Level	Colorimetric Methods							
		Froth Method (Saponins)	Gelatin-Salt Block Test (Tannins)	ABTS Assay (Antioxidant Activity)	Mayer’s Reagent (Alkaloids)	Salkowski’s Test (Steroids/ Terpenoids)	Folin–Ciocalteu Reagent (Phenols/Flavonoids)	Test Only for Phenols	Shinoda’s Reagent (Flavonoids)
*Daucus carota* (carrot)	crude extract	N.D.	N.D.	YES	YES	N.D.	YES	YES	N.D.
	purified active fractions	N.D.	N.D.	YES	N.D.	N.D.	N.D.	N.D.	N.D.
*Calendula* *officinalis* flowers	crude extract	N.D.	N.D.	YES	YES	YES	YES	N.D.	YES
	purified active fractions	N.D.	N.D.	YES	N.D.	N.D.	N.D.	N.D.	N.D.
*Aloe vera* leaves	crude extract	N.D.	N.D.	N.D.	YES	YES	YES	N.D.	YES
	purified active fractions	N.D.	N.D.	N.D.	YES	N.D.	YES	N.D.	YES

N.D. = not determined.

**Table 2 antioxidants-12-01242-t002:** Compounds identified by HR-MS/MS data (RDB = Ring and Double Bond; FA = Formate Anion; GPC = glycerophosphocholine; MGMG = monogalactosyl-monoacylglycerol; DGMG = digalactosyl-monoacylglycerol). Base peaks are reported in bold. The quantity in percentage is indicated.

Molecular ion (*m*/*z)*	Error (Ppm)	Formula	RDB	MS/MS Fragment Ions (*m*/*z*)	Tentative Assignment	*C.o.* (%)	*D.c.* (%)	*A.v.* (%)
269.0453	−0.9	C_15_H_10_O_5_	11	269.0458; 268.0379; 241.0510; 240.0428; 239.0342; 223.0394; 211.0396; 195.0444; 183.0447	Aloe-emodin			74.7
580.3288 [M + FA]^−^	5.5	C_27_H_52_NO_10_P	3	580.3291; 520.3071; 295.2278; 277.2175; 171.1026	GPC(OH18:2)	2.9		
559.3150 [M + FA]^−^	4.7	C_28_H_48_O_11_	5	513.3115; 277.2174; 253.0926	MGMG(18:3)	7.4	3.4	7.7
295.2279	0.1	C_18_H_32_O_3_	3	295.2281; 277.2173; 195.1396	Hydroxy-octadecadienoic acid	23.5	92.9	
699.3803 [M + FA]^−^	−0.8	C_32_H_60_O_16_	3	699.3882; 653.3805; 415.1470; 397.1363; 287.0772; 255.2334; 235.0826	DGMG(16:0)			17.6
562.3174 [M + FA]^−^	4.2	C_27_H_50_NO_9_P	4	562.3175; 502.2962; 277.2172; 224.0687	GPC(18:3)	13.7		
564.3320 [M + FA]^−^	2.3	C_27_H_52_NO_9_P	3	564.3338; 504.3114; 279.2326	GPC(18:2) isomer 1	4.2		
564.3323 [M + FA]^−^	2.3	C_27_H_52_NO_9_P	3	564.3345; 504.3124; 279.2336; 224.0693	GPC(18:2) isomer 2	29.7		
540.3307 [M + FA]^−^	0.0	C_25_H_52_NO_9_P	1	540.3337; 480.3107; 255.2325; 224.0663	GPC(16:0) isomer 1	2.2		
540.3323 [M + FA]^−^	3.0	C_25_H_52_NO_9_P	1	540.3347; 480.3116; 255.2327; 224.0689	GPC(16:0) isomer 2	13.9	3.6	
566.3484 [M + FA]^−^	2.0	C_27_H_54_NO_9_P	2	566.3485; 506.3269; 281.2477; 224.0691	GPC(18:1)	2.5		

*C.o.* = *Calendula officinalis*; *D.c.* = *Daucus carota*; *A.v.* = *Aloe vera*.

## Data Availability

The data are contained within the article and Appendix A.

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
