# Peer review of "Different Extraction Procedures Revealed the Anti-Proliferation Activity from Vegetable Semi-Purified Sources on Breast Cancer Cell Lines"

_antioxidants, 2023, doi:10.3390/antiox12061242_

Round 1

Reviewer 1 Report (Previous Reviewer 1)

This paper (revision of the previous paper antioxidants 2181525) describes how different extraction procedures revealed the anti-proliferation activity from vegetable semi-purified sources on breast cancer lines. The authors have made modifications to the previous manuscript, but I believe it still has the same limitations for publication in Antioxidants. The article is of scientific interest, but the authors still do not adequately describe the methods they use. Authors must take care of the formal aspects, fine-tune the scientific writing, extensively review the English writing, follow the journal template, etc. There has been some improvement in these areas, but they continue to be plagued by formal and formatting errors.

In addition to the comments described above, I consider that there are certain aspects that detract from the quality of the work:

- The materials and methods are not conveniently described or explained in detail.

- The authors continue only determine the existence or non-existence of certain families of compounds. It is true that the authors tentatively provide some compounds found in the extracts, but at no time do they quantify them, which detracts from the quality of the work. In this type of work, at least a total quantification of these families of compounds should be requested, although it would be interesting to have the quantification of at least the majority individual compounds.

Therefore, I continuo considering that without a more exhaustive characterization of the extract, this work does not have the necessary scientific relevance to be published in Antioxidants.

This paper (revision of the previous paper antioxidants 2181525) describes how different extraction procedures revealed the anti-proliferation activity from vegetable semi-purified sources on breast cancer lines. The authors have made modifications to the previous manuscript, but I believe it still has the same limitations for publication in Antioxidants. The article is of scientific interest, but the authors still do not adequately describe the methods they use. Authors must take care of the formal aspects, fine-tune the scientific writing, extensively review the English writing, follow the journal template, etc. There has been some improvement in these areas, but they continue to be plagued by formal and formatting errors.

In addition to the comments described above, I consider that there are certain aspects that detract from the quality of the work:

- The materials and methods are not conveniently described or explained in detail.

- The authors continue only determine the existence or non-existence of certain families of compounds. It is true that the authors tentatively provide some compounds found in the extracts, but at no time do they quantify them, which detracts from the quality of the work. In this type of work, at least a total quantification of these families of compounds should be requested, although it would be interesting to have the quantification of at least the majority individual compounds.

Therefore, I continuo considering that without a more exhaustive characterization of the extract, this work does not have the necessary scientific relevance to be published in Antioxidants.

Author Response

Referee 1 comment

This paper (revision of the previous paper antioxidants 2181525) describes how different extraction procedures revealed the anti-proliferation activity from vegetable semi-purified sources on breast cancer lines. The authors have made modifications to the previous manuscript, but I believe it still has the same limitations for publication in Antioxidants. The article is of scientific interest, but the authors still do not adequately describe the methods they use. Authors must take care of the formal aspects, fine-tune the scientific writing, extensively review the English writing, follow the journal template, etc. There has been some improvement in these areas, but they continue to be plagued by formal and formatting errors.

For the preparation of the manuscript, we followed the instructions and the journal template, the Editorial office did not reveal any problem about that. The English editing has been implemented through all the manuscript by Dr. Bernard Loeffler and Dr. Francesca Varrone.

In addition to the comments described above, I consider that there are certain aspects that detract from the quality of the work:

- The materials and methods are not conveniently described or explained in detail.

In the first and second step of revision, only the reviewer 3 (second step of revision) ask us some details about the crystal violet of cells staining. The judgment of referee 1 seem to be too generalized and it is not referred to a fact, on the contrary to what reviewer 3 requested. If there are paragraphs that are not clear we can modify them, but it is necessary for us to know what are these paragraphs and more important what is not clear.

- The authors continue only determine the existence or non-existence of certain families of compounds. It is true that the authors tentatively provide some compounds found in the extracts, but at no time do they quantify them, which detracts from the quality of the work. In this type of work, at least a total quantification of these families of compounds should be requested, although it would be interesting to have the quantification of at least the majority individual compounds.

During the first round of revision, we worked in collaboration with a research group from the University Vanvitelli in Caserta (Italy), in order to perform the mass spectroscopy analysis of the semi-purified extracts. Thus, we identified and quantized the compounds present in the extracts, as reported in table 2, and as indicated in the specific paragraph “UHPLC-HRMS and MS/MS analysis of semi-purified fractions” on page 3, lines 141-162.

Therefore, I continuo considering that without a more exhaustive characterization of the extract, this work does not have the necessary scientific relevance to be published in Antioxidants.

We are really disappointed about the referee 1 comments, since we have already answered all his/her request in the first revision step.

Reviewer 2 Report (Previous Reviewer 3)

All the comments and suggestions have been addressed. 

Author Response

Referee 2

All the comments and suggestions have been addressed. 

We thank referee 2 for his/her positive comments.

Reviewer 3 Report (New Reviewer)

In the paper entitled: “Different extraction procedures revealed the anti-proliferation activity from vegetable semi-purified sources on breast cancer cell lines,” the authors investigated the anti-proliferation potential of extracts from carrot, Calendula officinalis flower and Aloe vera on breast cancer vs epithelial cell lines.

Although well conducted, the work presents some issues to be addressed.

The Authors use Crystal Violet to check whether 1.0 % v/v DMSO final concentration affects the number and morphology of the examined cells (Figure 2). After how many hours they stained the cells? They should specify this point in MM and in the figure legend. In cell lines, i.e., MCF10A and MBA-MB-231, with different amounts of mitosis Crystal Violet may not be helpful. The different doubling times could affect the results of the treated cell lines. The authors should explain this point.

Moreover, it would be much more informative if the authors showed the crystal violet staining and morphology of cell lines treated with increasing doses of the various extracts in addition to the proliferation test with the different extracts.

The authors stated that the cell viability was measured after 24 and 48 hours of treatment, but they showed results only after 48 hours. Is it because 24 hours was too early to detect any effect on cell proliferation? Sometimes, 48 hours are insufficient to obtain any antiproliferative effect, and the best results are obtained after 72h. Did the author check this time point?

Authors should use the same spelling, "anti-proliferative or antiproliferative", throughout the manuscript.

The quality of English is good, although some mistyping errors should be corrected before publication.

i.e., Interesting the higher antiproliferative effect.. seem better to write "Interestingly" and more the one time.

And so on 

Author Response

Referee 3

In the paper entitled: “Different extraction procedures revealed the anti-proliferation activity from vegetable semi-purified sources on breast cancer cell lines,” the authors investigated the anti-proliferation potential of extracts from carrot, Calendula officinalis flower and Aloe vera on breast cancer vs epithelial cell lines.

Although well conducted, the work presents some issues to be addressed.

The Authors use Crystal Violet to check whether 1.0 % v/v DMSO final concentration affects the number and morphology of the examined cells (Figure 2). After how many hours they stained the cells? They should specify this point in MM and in the figure legend.

We thank referee 3 for his/her observation, in Fig.2, we stained the cells after 48 hours of growth in medium with and without DMSO 1%. We added this information in the manuscript as suggested.

In cell lines, i.e., MCF10A and MBA-MB-231, with different amounts of mitosis Crystal Violet may not be helpful. The different doubling times could affect the results of the treated cell lines. The authors should explain this point.

Moreover, it would be much more informative if the authors showed the crystal violet staining and morphology of cell lines treated with increasing doses of the various extracts in addition to the proliferation test with the different extracts.

We thank referee 3 for his/her comments, MCF10A and MBA-MB-231, have a different doubling times and in this case our purpose was to compare qualitatively the morphology and the number of the same cell line growing in presence or not of DMSO at 1%, by using this staining procedure, knowing the limits.

The authors stated that the cell viability was measured after 24 and 48 hours of treatment, but they showed results only after 48 hours. Is it because 24 hours was too early to detect any effect on cell proliferation? Sometimes, 48 hours are insufficient to obtain any antiproliferative effect, and the best results are obtained after 72h. Did the author check this time point?

We have measured the cell viability at 24, 48, 72 and 96 hours, and we have added this information in material and methods sections at 176-179 lines. We experienced for cell drugs we use in lab (i.e. dabrafenib, trametinib) the optimal time at 96 hours. However, the extracts we have used in this study, showed the best effect after 48 hours. 24 hours was too early. After 96 hours, cells started to proliferate again unless, after 48 hours of treatment, the medium was changed with fresh medium containing the tested extract. It is likely that after 48 hours the compound(s) from the extracts degrade or are metabolized within the cell. We are working on this issue as well as investigating the specific molecular and/or pathway target of the extract compound(s) we have identified in this study. This will be then the topic of another manuscript.

Authors should use the same spelling, "anti-proliferative or antiproliferative", throughout the manuscript.

We apologize for it, we corrected this error and we indicated all of them as “anti-proliferative”.

The quality of English is good, although some mistyping errors should be corrected before publication. i.e., Interesting the higher antiproliferative effect.. seem better to write "Interestingly" and more the one time. 

The English editing has been improved through all the text by Dr. Bernard Loeffler and Dr. Francesca Varrone.

Round 2

Reviewer 1 Report (Previous Reviewer 1)

The composition percentage (Table 2) could be accepted as valid for this preliminary study. Anyway I suggest small additional formal and format modifications prior to publication, that can be done during the proofreading of the manuscript.

Acceptable

Author Response

We thank referee 1 for her/his comment.

Reviewer 3 Report (New Reviewer)

The authors addressed most of the requests.

Is still required:

-check the English language 

-In the discussion, the authors state that they "investigated the anti-cancer properties.." which is not exactly what they did. They conducted a study on the effect of different substances on cancer cell lines' proliferation, which is different. For clearness, they should add the limitations related to the researcher in the discussion section. 

Checking the English language is required.

Author Response

The authors addressed most of the requests.

Is still required:

-check the English language 

-In the discussion, the authors state that they "investigated the anti-cancer properties.." which is not exactly what they did. They conducted a study on the effect of different substances on cancer cell lines' proliferation, which is different. For clearness, they should add the limitations related to the researcher in the discussion section. 

We agree with the referee, we corrected this last part of the manuscript, see page 11, lines 425-426; page 12, lines 431 and 436.

Comments on the Quality of English Language. Checking the English language is required.

We revised some parts of the manuscript; they are as new track change. The English editing was further revised by our editing assistants Bernard Loeffler and Francesca Varrone.

This manuscript is a resubmission of an earlier submission. The following is a list of the peer review reports and author responses from that submission.

Round 1

Reviewer 1 Report

This paper describes how different extraction procedures revealed the anti-proliferation activity from vegetable semi-purified sources on breast cancer lines. The article is of scientific interest, but the authors must describe in greater detail the methods they have used, as well as delve into the discussion of the results obtained. Authors must take care of the formal aspects, fine-tune the scientific writing, extensively review the English writing, follow the journal template, etc.

In addition to the comments described above, I consider that there are certain aspects that detract from the quality of the work:

- The materials and methods are not conveniently described or explained in detail.

- The authors only determine the existence or non-existence of certain families of compounds. In this type of work, at least a total quantification of these families of compounds should be requested, although it would be interesting to have the quantification of at least the majority individual compounds.

Therefore, I consider that without a more exhaustive characterization of the extract, this work does not have the necessary scientific relevance to be published in Antioxidants.

Author Response

We have added to the previously results the UHPLC-HRMS and MS/MS analysis of the semi purified fractions used in the biological assays. We have identified all the compounds present in the fractions and their compositions (see table 2), and based on these data we rewrote the “material”, “results” and “discussion” sections. Finally, the English editing was revised.

Reviewer 2 Report

This paper seems very intriguing, but its publication is approved if the authors added the chemical composition of the extracts (NMR or MS analyses) to relate the activity to the presence of some compounds able to modulate different pathways. Please add these experiments and write a good discussion.

Author Response

(The authors gave the same response as above.)

Reviewer 3 Report

The proposed study is investigating the anti-proliferation potential of extracts from carrot, Calendula officinalis flower, and Aloe vera on breast cancer vs. epithelial cell lines by using different extraction procedures.

I have some suggestions to improve the quality of the paper:

  1. Please add the literature review or related works from previous studies that are related to the proposed study.
  2. The results could be compared to previous studies (if available).
  3. Please add the conclusion section to the paper, including the limitations of the proposed study and challenges for future works.

Author Response

We have added the UHPLC-HRMS and MS/MS analysis of the semi purified fractions used in the biological assays. We have identified all the compounds presents and based on these results we have compared our data with studies previously reported in literature.

As request by the referee, we added a “conclusion” section.